# Prenatal Ultrasound Findings and Chromosomal Outcomes of Pregnancies with Mosaic Embryo Transfer

**DOI:** 10.3390/diagnostics14242795

**Published:** 2024-12-12

**Authors:** You Mi Hong, Soo Hyun Kim, Hee Jin Park, Hyun Mee Ryu, Dong Hyun Cha, Moon Young Kim, You Jung Han

**Affiliations:** 1Department of Obstetrics and Gynecology, CHA Gangnam Medical Center, CHA University School of Medicine, Seoul 06125, Republic of Korea; hong4136@chamc.co.kr (Y.M.H.); soohyunkim@chamc.co.kr (S.H.K.); coolsome72@chamc.co.kr (H.J.P.); chadh001@chamc.co.kr (D.H.C.); mykimdr410@chamc.co.kr (M.Y.K.); 2Department of Obstetrics and Gynecology, CHA Bundang Medical Center, CHA University School of Medicine, Seongnam 13496, Republic of Korea; hmryu@cha.ac.kr

**Keywords:** mosaic embryo transfer, pregnancy outcome, preimplantation genetic testing for aneuploidy, prenatal genetic diagnosis, prenatal ultrasound

## Abstract

Background: To investigate prenatal ultrasound findings and the chromosomal outcomes of mosaic embryo transfer. Methods: This retrospective study was conducted on pregnant women who underwent mosaic embryo transfer following blastocyst-stage preimplantation genetic testing for aneuploidy (PGT-A) at CHA Gangnam Medical Center from January 2021 to July 2024. Trophectoderm biopsy specimens were collected using standard protocols, and next-generation sequencing profiles were defined as mosaics when displaying copy number counts in the 20–80% range. The results of the PGT-A, the amniocentesis results, the findings of prenatal ultrasounds, and the pregnancy outcomes were analyzed. Results: A total of 88 mosaic embryos were transferred, of which 77 embryos were successfully implanted. Sixty-seven embryo-maintained pregnancies went beyond 11 weeks (87.0%), all among 58 patients with singleton pregnancies. The chaotic subtype showed the lowest ongoing pregnancy rate, and high-level mosaicism was less frequent in the ongoing group, compared to the total study group and the successful implantation group. Amniocentesis was performed on 33 mothers (56.9%), revealing two cases with abnormal findings that did not correlate with the PGT-A results. Two cases showed abnormalities in the second trimester detailed ultrasound, and both subsequently demonstrated normal findings in the third trimester and after birth. The average gestational age at birth was 38.4 weeks, and the average birth weight was 3313 g. No congenital anomalies were detected in 16 postnatal cases. Conclusions: Our study indicated that mosaic embryos can develop into euploid healthy infants with various levels or types of mosaicism, although the postnatal follow-up data are limited. This study is invaluable for counseling clinical results after mosaic embryo transfer, reassuring that, if patients do not have euploid embryos available, mosaic embryos can also be a viable option for transfer.

## 1. Introduction

As the age of mothers is increasing worldwide, the use of in vitro fertilization (IVF) is also on the rise. To improve IVF transfer outcomes, chromosomal testing, such as preimplantation genetic testing for aneuploidy (PGT-A), has been applied in clinical practice. Preimplantation genetic testing (PGT) involves examining the blastocyst, a day-5 embryo, to diagnose chromosomal abnormalities, allowing for the selection of embryos before implantation [1]. PGT can be further categorized based on the type of chromosomal abnormalities being tested, including PGT-A, PGT-M (monogenic/single-gene disorders), and PGT-SR (structural rearrangements). Among these, PGT-A focuses specifically on aneuploidy screening. As the age of mothers undergoing IVF increases, the frequency of aneuploidy findings in PGT-A results also rises. PGT-A can be used not only in cases of recurrent miscarriage or a history of chromosomal abnormality pregnancies but also to select and implant only euploid embryos to increase the chances of a successful pregnancy [1,2].

PGT has rapidly advanced due to developments in genetic technology and improvements in the IVF laboratory environment. With the introduction of NGS, previously undetectable mosaic embryos have been identified. Several studies showed that the incidence of mosaic embryos in PGT-A with NGS profiling was 2–40% [3,4,5,6].

Embryo mosaicism can be defined as the existence of two or more cell lines in an embryo [7]. Several recent studies have reported on mothers who underwent mosaic embryo transfer and showed lower implantation rates and higher miscarriage rates compared to those who received transfers of normal embryos [8,9,10]. However, based on the current results, mosaic embryos should not be systematically discarded or excluded from transfer [6]. A mosaic embryo transfer becomes a viable alternative when no euploid embryos are available for transfer from the PGT-A results [7,11]. Therefore, various societies, including the Preimplantation Genetic Diagnosis International Society (PGDIS) and the American Society for Reproductive Medicine (ASRM), offer recommendations regarding the transfer of mosaic embryos in general [5,11].

Even when pregnancy is achieved after the transfer of a mosaic embryo, there are concerns about potential chromosomal abnormalities in the fetus, congenital malformations, and long-term complications in the newborn due to the remaining abnormal cells. Currently, only a limited number of case studies have been published on these outcomes [9,12].

Therefore, this study aims to investigate the pregnancy success rates of mosaic embryo transfer, the prenatal ultrasound findings, and the chromosomal test results of the mothers who conceived through this method. Additionally, it seeks to examine the delivery outcomes and the presence of congenital anomalies in newborns.

## 2. Materials and Methods

### 2.1. Study Design

This retrospective cohort study consists of cases at CHA Gangnam Medical Center, where the mothers who underwent PGT-A also received mosaic embryo transfers, covering cases from January 2021 to July 2024. In the study, the indications for PGT-A include advanced maternal age (typically over 35 years), repeated implantation failure, history of recurrent miscarriages, previous pregnancy with chromosomal abnormalities, and couples desiring to increase the chance of a successful pregnancy. The data were retrospectively collected by reviewing the electronic medical records related to IVF and PGT procedures, as well as the records of mothers and their newborns. The study was approved by the institutional review board at CHA Gangnam Medical Center (IRB number: 2021-12-002-008).

### 2.2. PGT-A Methods

All embryos in this study underwent blastocyst-stage PGT-A using the same NGS-based platform VeriSeq (Illumina, CA, USA) after whole-genome amplification with the Sureplex WGA kit (Illumina, CA, USA) and subsequent frozen embryo transfer. Trophectoderm biopsy specimens were collected using standard protocols and frozen until processing. Mosaicism is defined as two or more cell populations with different genotypes. NGS profiles were defined as mosaic when displaying copy number counts in the 20–80% range. Profiles < 20% were considered to be euploid, and those >80% were considered to be aneuploid [9,12,13]. The resolution of PGT-A is 5–10 Mb.

### 2.3. Mosaic Embryo Classification

Mosaic embryos were classified based on three criteria. First, the number of abnormal chromosomes was categorized as follows: “simple” for one abnormal chromosome, “complex” for two to three abnormal chromosomes, and “chaotic” for four or more abnormal chromosomes. Second, the proportion of abnormal cells was assessed, with “low level” indicating an intermediate copy number present in 20–40% of the cells, and “high level” indicating an intermediate copy number present in 40–80% of the cells [14,15]. Finally, the type of chromosomal aberration was determined, where “whole type” referred to an intermediate copy number distributed throughout the entire chromosome, while “segmental type” indicated that the intermediate copy number was present only in specific segments of the chromosome [8].

### 2.4. Genetic Counseling

Before the transfer of mosaic embryos, clinicians provided genetic counseling, including the alternative options, potential risks, limited neonatal outcomes, and prenatal management of mosaic embryo transfers. After pregnancy was confirmed, prenatal genetic counseling was provided. Amniocentesis was recommended to diagnose a euploid fetus, but prenatal diagnostic testing was performed with the test preferred by the patient. If prenatal diagnostic testing is performed, additional analyses, such as chromosomal microarray (CMA) or uniparental disomy (UPD) testing, beyond karyotyping, were also considered based on the specific PGT-A results [11,15].

### 2.5. Clinical Outcomes

The study population was divided into three groups for analysis: (i) the overall group consisting of all mothers who received mosaic embryo transfers, (ii) the successful implantation group where an intrauterine gestational sac was observed, and (iii) the ongoing pregnancy group where pregnancies were sustained for more than 11 gestational weeks. Demographic characteristics and mosaic subtypes were compared across these groups. For the participants who achieved pregnancy, further analysis was conducted on the prenatal ultrasound findings, which included a first-trimester nuchal translucency scan and a second-trimester detailed ultrasound, prenatal genetic diagnosis, and obstetric and neonatal complications.

### 2.6. Statistical Analysis

In this study, categorical variables and continuous variables with normal and non-normal distribution were expressed as *n* (%), mean standard deviation (SD), and median, respectively. The ANOVA test and Chi-square test were used to analyze the continuous and categorical variables, respectively. The data analyses were performed using the Statistical Package for Social Sciences version 26.0 (IBM Corp., Chicago, IL, USA), and *p* < 0.05 denoted statistical significance.

## 3. Results

### 3.1. Study Population

During the study period, a total of 88 mosaic embryos were transferred, and in 23 cases, two embryos were implanted. Of the 77 successfully implanted embryos, 67 embryos maintained ongoing pregnancy beyond 11 gestational weeks without miscarriage, yielding a rate of 87.0% (67/77). In the Ongoing Pregnancy group, while 67 embryos were transferred among 58 patients, all resulted in singleton pregnancies, with one embryo implanting per patient. Detailed prenatal ultrasounds were performed in 43 cases by the first trimester and in 35 cases by the second trimester. Amniocentesis was performed in 33 cases. Cases were excluded where prenatal tests had not yet been performed or when the patient was transferred to another hospital during the early stages of pregnancy (Appendix A).

### 3.2. Baseline Characteristics

In the comparison of the total study group, the successful implantation group, and the ongoing pregnancy group, the ongoing pregnancy group had the lowest average age at 38.2 years, with the lowest proportions of mothers aged 40 years and above, as well as those aged 45 years and above. However, these differences between the three groups were not statistically significant. Regarding the history of miscarriage, all three groups exhibited similar frequencies, with approximately 55–65% (49/88, 45/77, 43/67) of participants having a history of miscarriage (Table 1).

### 3.3. Comparison of Mosaic Embryo Subtypes

In the comparison of mosaic embryo subtypes, the chaotic type, characterized by four or more abnormal chromosomes, was found to be the lowest in the ongoing pregnancy group at 20.9% (14/67). Additionally, the proportion of high-level mosaicism was also lower in the ongoing pregnancy group compared to the other two groups. However, the distribution of these mosaic embryo subtypes did not show statistically significant differences between the three groups. There were no differences in the frequencies of whole type and segmental type among the three groups (Table 2).

### 3.4. Prenatal Genetic Diagnosis

Amniocentesis was performed in 33 cases (56.9%, 33/58). Among these, conventional karyotyping was conducted in 32 cases, and CMA was conducted in 10 cases. Twenty-three cases involved only conventional karyotyping, one case involved only CMA, and nine cases involved both tests. NIPT (non-invasive prenatal testing) was performed in 6 cases (10.3%, 6/58), with no abnormal results detected. Amniocentesis was not performed in any of the six cases where NIPT was conducted. Since this is a standard NIPT, it is not suitable for detecting specific mosaic abnormalities. There were only two cases that showed abnormal findings for the invasive tests. In one case, conventional karyotyping revealed a translocation between chromosomes 2 and 6. The mosaic embryo subtype for this fetus involved chromosomes 3 and 16, indicating no correlation between the PGT-A results of the transferred mosaic embryo and the amniocentesis results. A subsequent parental karyotype analysis showed the same translocation in the mother. Additionally, one case showed a 1.5 Mb gain in the Xp11.21 region on the CMA using the reference genome version GRCh37. The general minimum resolution of the CMA (cytoscan optima assays) is 1 Mb for deletions, 2 Mb for duplications, and 5 Mb for loss of heterozygosity. For approximately 400 critical genes related to prenatal genetic disorders, the copy number variation detection resolution is up to 100 Kb, allowing for the detection of a 1.5 Mb amplification on the X chromosome. It was classified as a VOUS (variant of uncertain significance) according to the five tiers of the American College of Medical Genetics and Genomics (ACMG) classification. For this fetus, the mosaic embryo subtype involved chromosomes 3, 16, and 2, similarly showing no correlation between the PGT-A results of the transferred mosaic embryo and the amniocentesis findings. We recommended parental karyotyping to the patient regarding the amniocentesis results, but the patient’s guardian declined. So, we were unable to confirm whether the chromosomal abnormality originated from the parents (Table 3).

### 3.5. Prenatal Ultrasound Findings

Detailed prenatal ultrasounds were performed in 43 cases (74.1%, 43/58) by the first trimester and in 35 cases (60.3%, 35/58) by the second trimester. No abnormalities were detected in the first-trimester nuchal translucency scan, while two cases showed abnormalities in the second-trimester detailed ultrasound. The two cases involved transfers of single, low-level, whole-type mosaic embryos of chromosomes 9 and 1, respectively (Table 3). Both showed mild renal pelvis dilatation in the second trimester, but one case resolved to normal findings in the third trimester, and the other was confirmed to be normal after birth (Figure 1). Both cases that showed abnormal findings on the prenatal ultrasound underwent amniocentesis, and both cases had normal karyotype and chromosomal microarray analysis results.

### 3.6. Perinatal Outcomes

Among the 58 mothers in the ongoing pregnancy group, excluding those who were still pregnant or transferred to another hospital, a total of 16 mothers, accounting for 27.6% (16/58) of the group, provided perinatal data after giving birth at our institution (Table 4). The average gestational age at birth was 38.4 weeks, and the average birth weight was 3313 g. Neonatal intensive care unit (NICU) admission was required in three cases (18.7%, 3/16), with an average stay of 8 days. Neonatal complications included transient tachypnea of the newborn in two cases and neonatal jaundice in one case. Although the number of cases is small, there was one case of preterm birth at 36 weeks, but no congenital anomalies were detected in any of the newborns. An analysis of the mosaic subtypes of the embryos transferred in these 16 patients revealed that, compared to complex types, single types were more frequently observed, with segmental types being more common than whole types and low-level mosaic types more prevalent than high-level mosaic types (Table 4).

## 4. Discussion

We reported an ongoing pregnancy rate of 76.1% (67/88) for mosaic embryo transfers. Among these cases, no significant chromosomal abnormalities or anomalous prenatal ultrasound findings were observed, and there were no instances of congenital anomalies or complications after birth.

Despite the increasing sensitivity of newer PGT-A platforms, it remains to be seen whether a single trophectoderm biopsy is truly representative of the entire embryo. It is important to note that PGT involves performing a biopsy not on the inner cell mass, which develops into the fetus, but on a portion of the trophectoderm, which develops into the placenta. As a result, the test outcome can vary depending on which specific area of the trophectoderm is sampled [2]. Additionally, the varying criteria for reporting mosaicism across laboratories, along with the recent surge in NGS-based resolution, not only reveal more abnormal segmental mosaicism but also lead to an increase in false-positive results [16,17].

In our study, the majority of the 33 amniocentesis results were normal, with abnormalities found in only two cases [18,19]. However, these abnormalities were completely different chromosomal issues unrelated to the mosaicism results of the PGT-A and were not clinically significant. Because the mosaicism at the blastocyst stage and during gestation did not match, they were likely unrelated. Probably, the original embryonic mosaicism resolved itself (self-correction), although a later mitotic error established the mosaicism observed with the prenatal test. Such occurrences are to be expected, considering that 0.25–0.4% of all amniocenteses indicate mosaicism [13]. So far, only two cases have been reported with postnatal confirmation of mosaicism, with one following the transfer of a low-range mosaic embryo and the other after the transfer of a high-level mosaic embryo with trisomy 15 and partial monosomy 20 [18,19].

Although the frequency of mosaic embryo transfers has also increased dramatically with the rise in PGT in recent years (Figure 2), recent studies have indicated that mothers undergoing mosaic embryo transfers experience lower implantation rates and higher miscarriage rates than those receiving normal embryo transfers [6,20,21,22,23]. Zhang et al. reported that mosaic embryo transfers resulted in a significantly lower clinical pregnancy rate (40.1% versus 59.0% versus 48.4%), a lower ongoing/live birth rate (27.1% versus 47.0% versus 35.1%), and a higher miscarriage rate (33.3% versus 20.5% versus 27.4%) than euploid and non-PGT transfer, respectively [24]. We reported that, among 88 cases of mosaic embryo transfers, 67 embryos maintained an ongoing pregnancy beyond 11 weeks, resulting in a relatively high ongoing pregnancy rate of 76.1%. However, we could not compare these results with those from the euploid embryo transfers conducted during the same period.

To date, most studies have reported live birth rates up to the point of pregnancy, with very few compared to prenatal chromosomal testing results. Zhang et al. reported that, out of 102 patients who underwent mosaic embryo transfer, 4 patients underwent amniocentesis after mosaic embryo transfer, and no chromosomal abnormalities were detected among 48 newborns [24]. Victor also reported results from 11 cases where amniocentesis was performed out of 100 mosaic embryo transfers, noting that one case showed a balanced translocation, and two cases had microdeletions smaller than the valid resolution of the PGT-A platform [25]. Our study involved amniocentesis in a relatively high proportion of mothers who underwent mosaic embryo transfer (56.9%, 33/58), and we report the results. Abnormal findings in amniocentesis were very few (6.1%, 2/33 cases), and even in cases with abnormal findings, these were unrelated to the PGT-A results of the transferred embryos.

Although there are not many cases of newborns born from mosaic embryo transfers, our study found that none of the 16 newborns exhibited congenital anomalies after birth. Due to the recent increase in mosaic embryo transfers, other studies have primarily focused on prenatal outcomes, resulting in limited reports on newborns [9,10]. A recently published multicenter study reported that, among 488 newborns from mosaic embryo transfers, only one case of major congenital anomaly was found, which is lower than the 2–3% reported for fetuses born through assisted reproductive technology [26].

There is no consensus regarding specific mosaic features and their impact on treatment and pregnancy outcomes [10,14,16,24,25]. Vitor et al. analyzed single segmental types and multiple segmental types, finding that the implantation rate (45.5% vs. 36.4%) and clinical pregnancy rate (39.4% vs. 27.3%) were higher in single types [25]. Spinella et al. reported results based on mosaicism ratios, finding that mosaic embryos with <50% mosaicism resulted in higher implantation rates (48.9% vs. 24.2%), as well as improved clinical pregnancy outcomes (40.9% vs. 15.2%) and live birth rates (42.2% vs. 15.2%) compared to those with >50% mosaicism [27]. Munne et al., comparing clinical outcomes after the transfer of mosaic embryos with segmental mosaicism and those with single whole chromosome mosaicism, did not find a significant difference [28]. In our study, similarly, the chaotic subtype exhibited the lowest ongoing pregnancy rate, and high-level mosaicism was also less frequent in the ongoing group compared to both the total study group and the successful implantation group. In the analysis of delivery information from the 16 cases, single types were observed more frequently than complex types, with segmental types being more common than whole types, and low-level mosaic types being more prevalent than high-level mosaic types.

Most guidelines, including those from the American Society for Reproductive Medicine, indicate that the primary advantage of prenatal testing following mosaic embryo transfer is amniocentesis [11,12,15]. Although chorionic villus sampling was performed earlier, there are limitations to analyzing cells that are placental in origin, similar to PGT-A which tests only trophectoderm DNA (deoxyribonucleic acid). Because cell-free DNA testing, NIPT, also analyzes the free-floating placental DNA present in maternal blood, it is not designed for the detection of mosaicism and may result in false-negative or positive results [8,11,12]. In our study, no cases of chorionic villus sampling were performed, and among the total number of mothers, six (10.3%, 6/58) chose NIPT instead of amniocentesis. Additionally, the current guidelines emphasize that patients must be informed that amniocentesis may miss low-level mosaicism or non-placental, non-epithelial cell mosaicism if selected. Furthermore, when conducting the test, it should be accompanied by considerations based on PGT-A results, such as chromosomal microarray analysis, UPD, and additional cell counts [8,11,12,15,29].

The primary strength of our study is that, compared to several other studies on patients with mosaic embryo transfers, we performed and reported amniocentesis and detailed prenatal ultrasounds in approximately 50–60% (33/58, 35/58) of participants, a relatively high proportion, and conducted a comparative analysis with the PGT-A results of the transferred mosaic embryos. Additionally, our study used PGT-A results from a single laboratory within a single institution, ensuring consistent criteria. This allowed for a sequential analysis from embryo transfer through to the neonatal stage. Despite a number of strengths, our study has some limitations. A significant number of patients had recently undergone mosaic embryo transfer, so postnatal information (16/58 cases) was limited as many had not yet given birth. Although the cases showing abnormal findings in both amniocentesis and detailed ultrasound were very few (2/33 and 2/35 cases, respectively), it cannot be ruled out that negative outcomes may have occurred in the missing postnatal follow-up data. When two embryos were transferred to a single mother, only one implanted. But, both may have been counted as successful, leading to an overestimation of implantation and pregnancy success rates. As a limitation of the study, we were unable to perform a placenta biopsy after delivery in mosaic embryo transfer pregnancies, and thus, postnatal confirmation could not be carried out. Therefore, there is a possibility that cases could result in confined placental mosaicism which requires specific obstetric management due to the risk of reduced fetal growth and hypertensive disorders [30]. We plan to conduct a longitudinal analysis of the large-scale data collected from our ongoing mosaic embryo transfer pregnancy cases, starting from the embryo stage and including the prenatal, postnatal, and particularly long-term outcomes of infants born through mosaic embryo transfer.

## 5. Conclusions

Our study results suggest that mosaic embryos can also lead to euploid pregnancies and healthy live births. This may provide valuable data that support mosaic embryo transfer as a viable option for patients who are unable to undergo transfers of euploid embryos.

## Figures and Tables

**Figure 1 diagnostics-14-02795-f001:**
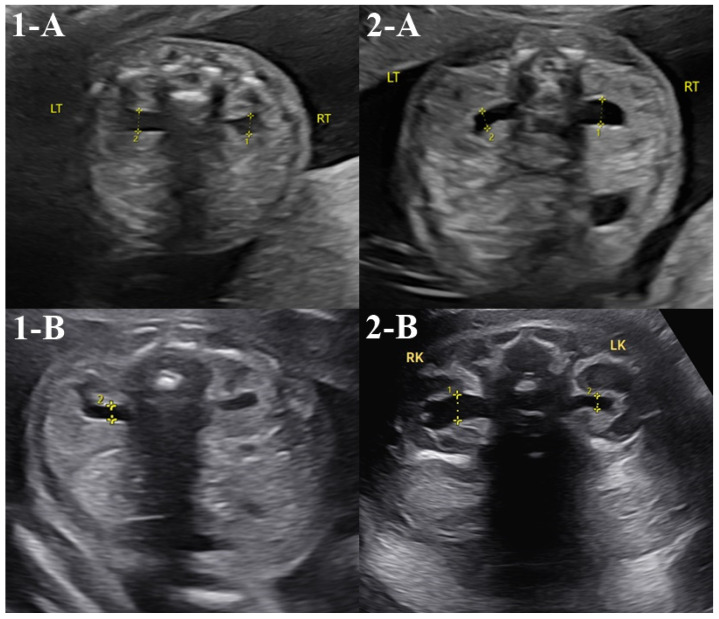
Abnormal prenatal detailed ultrasound cases. Asterisk means anteroposterior renal pelvic diameter. In the second-trimester detailed ultrasound, two cases exhibited abnormal findings. Case 1 showed mild bilateral renal pelvis dilation on the detailed ultrasound performed at 22 weeks (**1**-**A**, anteroposterior renal pelvic diameter left 4.0mm, right 4.5 mm). However, follow-up examinations at 24 weeks confirmed normal findings bilaterally (**1**-**B**). Case 2 demonstrated mild right renal pelvis dilation on the detailed ultrasound performed at 21 weeks (**2**-**A**, anteroposterior renal pelvic diameter right 4.3 mm), and the same findings were noted on another detailed ultrasound at 36 weeks (**2**-**B,** anteroposterior renal pelvic diameter right 8.0 mm). However, postnatal examination confirmed normal findings bilaterally in the newborn.

**Figure 2 diagnostics-14-02795-f002:**
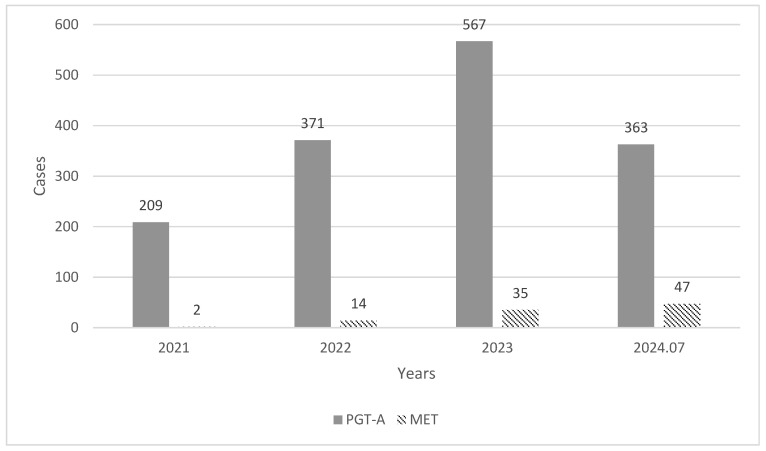
Annual trend in PGT-A and MET cases in our center. PGT-A preimplantation genetic testing for aneuploidy; MET mosaic embryo transfer. The figure shows a significant upward trend in the number of PGT-A (solid bars) and MET (striped bars) cases from 2021 to 2024. PGT-A cases increased from 209 in 2021 to 363 by July 2024, more than tripling. Similarly, MET cases rose sharply from 2 in 2021 to 47 by 2024, indicating a more than 40-fold increase.

**Table 1 diagnostics-14-02795-t001:** Baseline characteristics in three groups.

	Total Study Group *n* = 78	Successful Implantation Group *n* = 67	Ongoing Pregnancy Group *n* = 58	*p*
No. of embryos	88	77	67	
Average maternal age (years)	39.1 ± 3.2	38.7 ± 3.1	38.2 ± 3.0	0.487
>35 years (%)	85 (96.6)	74 (96.1)	65 (97.0)	0.801
>40 years (%)	42 (47.7)	31 (40.2)	25 (37.3)	0.482
>45 years (%)	4 (4.5)	2 (2.6)	1 (1.5)	0.527
Previous miscarriage history				0.933
No	39 (44.3)	32 (41.5)	24 (35.8)	
1 experience (%)	25 (25.0)	21 (27.3)	20 (29.8)	
>2 experience (%)	27 (30.7)	24 (31.2)	23 (34.3)	

**Table 2 diagnostics-14-02795-t002:** Mosaic embryo subtypes in three groups.

	Total Study Group *n* = 78	Successful Implantation Group *n* = 67	Ongoing Pregnancy Group *n* = 58	*p*
No. of embryos	88	77	67	
Number, *n* (%)				
Single	39 (44.3)	36 (46.7)	31 (46.3)	0.988
Complex	28 (31.8)	23 (29.9)	22 (32.8)	
Chaotic	21 (23.9)	18 (23.4)	14 (20.9)	
Proportion, *n* (%)				0.864
Low level	78 (88.6)	70 (90.9)	61 (91.0)	
High level	10 (11.4)	7 (9.1)	6 (9.0)	
Type, *n* (%)				0.984
Whole type	53 (60.2)	46 (59.7)	41 (61.2)	
Segmental type	35 (39.8)	31 (40.2)	26 (38.8)	

**Table 3 diagnostics-14-02795-t003:** Comparison of abnormal prenatal test results with transferred embryo PGT results in the ongoing pregnancy group.

	Abnormal Results on Prenatal Tests	Mosaic Embryo Details on PGT
Amniocentesis		
Karyotype (%)	3.1%, 1/32	
	46,XX,t(2;6)(q36;q24), mat	Whole mos: Chr3 (+20%), Chr16 (+20%)
CMA (%)	10%, 1/10	
	Arr[GRCh37] Xp11.21(55489634_57022462) X 3 1.5Mb Gain VOUS	Whole mos: Chr3 (+20%), Chr16 (+20%), Seg mos: Chr2 (q22.1–q35) (+30%)
Detailed ultrasound		
1st trimester (%)	0%, 0/43	
2nd trimester (%)	5.7%, 2/35	
	Mild renal pelvis dilatation	Whole mos: Chr9 (−20%)
	Mild renal pelvis dilatation	Whole mos: Chr1 (+20%)

PGT, preimplantation genetic testing; CMA, chromosomal microarray analysis.

**Table 4 diagnostics-14-02795-t004:** Details of mosaic embryos in delivered patients.

	PGT-A Results
Case 1	Whole mos: Chr3 (−20%), Chr22 (−20%), Seg mos: Chr2 (+20%), Chr14 (+30%), Chr18 (−20%), Chr21 (+20%)
Case 2	Seg mos: Chr5 (+30%)
Case 3	Whole mos: Chr1 (+20%)
Case 4	Seg mos: Chr9 (−30%)
Case 5	Seg mos: Chr8 (+20%), Chr12 (+30%)
Case 6	Whole mos: Chr9 (−20%), Seg mos: Chr11 (+30%)
Case 7	Seg mos: Chr2 (−30%)
Case 8	Whole mos: ChrX (+20%)
Case 9	Whole mos: Chr2 (+30%), Chr11 (+30%), Chr12 (−20%), Chr16 (−20%), Chr18 (+50%), Chr21 (+20%)
Case 10	Whole mos: Chr10 (+20%), Chr20 (+20%)
Case 11	Whole mos: Chr1 (+30%), Chr8 (−40%)
Case 12	Seg mos: Chr14 (+30%)
Case 13	Whole mos: Chr6 (+20%)
Case 14	Seg mos: Chr2 (−20%), Chr5 (−40%), Chr16 (+40%)
Case 15	Whole mos: Chr1 (+30%)
Case 16	Whole mos: Chr19 (+60%)

## Data Availability

The detailed data presented in this study are available from the corresponding author upon request.

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
