# Peer review of "Prenatal Ultrasound Findings and Chromosomal Outcomes of Pregnancies with Mosaic Embryo Transfer"

_diagnostics, 2024, doi:10.3390/diagnostics14242795_

Round 1
Reviewer 1 Report
Comments and Suggestions for Authors
Thank you for this interesting study, which I gladly reviewed. Let me give you some suggestions, which could improve your paper.
First of all, for the future, please include line numbers in your text, as it is far easier to communicate with your reviewer. By the lack of it, I will give you page numbers now.
p2: 'With the application of next generation sequencing (NGS), which has high sensitivity and resolution, to PGT-A, there has been an increase in embryonic mosaicism recently' => the mosaic embryo's where always there, we just didn't have the means to detect them before; please correct.
p3: 'Profiles <20% were considered euploid, and those >80% were considered aneuploidy' => either use adjectives or substantive, but be consistent.
p5 table 1/2: '*p <0.05, **p < 0.005.' => regarding the results, subtext is unnecessary
p6 'The mosaic embryo subtype for this fetus involved chromosomes 3 and 16, indicating no correlation between the PGT-A results of the transferred mosaic embryo and the amniocentesis results. A subsequent parental karyotype analysis showed the same translocation in the mother.'=> a karyotyping of both partners is standard procedure before accepting a couple for IVF-PGT, how come this was not performed?
p6: parental origin of the 1.5 Mb duplication on the X chromosome?
p6 'Prenatal ultrasound findings'=> karyotype of CMA results of the two cases?
p9 'So far, only two cases have been reported showing postnatal confirmation of mosaicism, one after the transfer of a low-range (35%) mosaicism, one after the transfer of a low-range (35%) mosaicism of monosomy 2 and the other after the transfer of high-level mosaic for trisomy 15 and partial monosomy 20.' => this sentence is linguistically incorrect, please correct
p10 'Although chorionic villus sampling is an earlier,'=> same remark
Comments on the Quality of English Language
Please ask a native speaker to read and adjust your text.
Reviewer 2 Report
Comments and Suggestions for Authors
General Comments:
This manuscript presents valuable data on chromosomal outcomes and prenatal ultrasound findings following mosaic embryo transfer, with a particularly noteworthy contribution being the substantial number of amniocentesis results. The study's focus on genetic testing outcomes after PGT-A is relatively unique in the current literature. However, there are several significant concerns regarding data presentation, consistency, and the strength of conclusions that need to be addressed before publication.
Major Concerns:
1. Data Presentation and Consistency:
- There are multiple discrepancies between numbers presented in the text and tables:
* The total number of embryos (88) doesn't align with the case breakdown (23 cases with two embryos [46] plus single embryo cases)
* Table 1 shows "Total Study group N=77" which doesn't match the total case calculation
* Similar inconsistencies exist for the "Successful implantation group" and "Ongoing pregnancy group"
- A flow chart or visual representation would significantly improve clarity of the study population and outcomes
2. Study Outcomes and Conclusions:
- The conclusion that "mosaic embryos can develop into euploid healthy infants" is based on only 16 postnatal cases out of 58 ongoing pregnancies
- The manuscript should explicitly address the limitations of missing outcome data for 42 cases (72.4%) of the ongoing pregnancy group
- This substantial loss to follow-up significantly weakens the strength of the conclusions
3. Methodological Clarity:
- The definition of mosaicism is inconsistent:
* Methods section 2-2 defines mosaic range as 20-80%
* Section 2-3 describes low level as 30-50% and high level as 50-70%
* This discrepancy needs resolution and explanation
Specific Recommendations:
1. Abstract:
- Correct spelling: "healty" to "healthy"
- Add a clear statement about the correlation (or lack thereof) between PGT-A detected mosaicism and amniocentesis findings
- Revise conclusions to reflect the limited postnatal follow-up data
2. Methods:
- Clarify mosaicism definitions and ranges
- For amniocentesis data (33 cases, 56.9%):
* Specify the denominator used for percentage calculations
* Clarify the overlap between conventional karyotyping (32 cases) and CMA (10 cases)
* Explain how NIPT cases (6) relate to the amniocentesis group
3. Results:
- Correct section numbering (currently duplicate 3-3)
- Regarding NIPT:
* Specify whether genome-wide NIPT or standard NIPT was used
* Clarify its appropriateness for detecting the specific mosaic abnormalities
- For CMA findings:
* Add reference genome version (GRCh37 or GRCh38)
* Explain that the 1.5 Mb gain on X chromosome was below PGT-A resolution
- Clarify maternal age reporting (average vs. median) between text and Table 1
4. Discussion:
- Add a comprehensive discussion of study limitations, particularly:
* Limited postnatal follow-up (16/58 cases)
* Potential impact of missing data on conclusions
- Strengthen discussion of the significance of amniocentesis findings relative to PGT-A results
Technical Corrections:
1. Change "NSG" to "NGS" throughout
2. Correct all statistical calculations with clear denominators
3. Ensure consistent numbering of sections
4. Add reference genome version for array results
The manuscript presents potentially valuable data, particularly regarding amniocentesis outcomes after mosaic embryo transfer. However, substantial revision is needed to address the identified issues, particularly regarding data presentation and the strength of conclusions given the limited follow-up data. A more measured conclusion acknowledging these limitations would strengthen the paper's contribution to the field.
Round 2
Reviewer 2 Report
Comments and Suggestions for Authors
I have reviewed the revised manuscript and noted several improvements from the previous version. However, there are still some important issues that need to be addressed:
- Data Consistency Issues
Line 177 There appears to be a discrepancy between "Of all 58 patients were singleton pregnancies" and the number of embryos (67) shown in Table 1 for the Ongoing Pregnancy group (N=58). Shouldn't the number of embryos be 58 if these were all singleton pregnancies? Perhaps this means that 67 embryos were transferred among 58 patients, but all resulted in singleton pregnancies. If so, this should be clearly stated in the text: "In the Ongoing Pregnancy group, while 67 embryos were transferred among 58 patients, all resulted in singleton pregnancies with one embryo implanting per patient." This clarification should also be reflected in Table 1 and Supplementary Figure 1.
Line 187 The text mentions "lowest average age at 38.2 years" but this specific value does not appear in Table 1, creating an inconsistency.
- Technical Details
Line 227 After "version GRCh37," the manuscript should include information about the general resolution of PGT-A to explain why this 1.5 Mb gain could not be detected.
Lines 243-244 In Table 3, "arr Xp11.21(55489634_57022462)X3" should be modified to "arr[GRCh37] Xp11.21(55489634_57022462)X3" for technical accuracy.
- Structural Issues
Section Numbering
- Line 245: "3-4. Prenatal ultrasound findings" should be "3-5. Prenatal ultrasound findings"
- Line 269: "3-5. Perinatal outcomes" should be "3-6. Perinatal outcomes"
Statistical Correction Line 392: "six (10.3%, 16/58)" should be corrected to "six (10.3%, 6/58)"
- Supplementary Figure 1
Multiple numerical inconsistencies exist in the figure. Specifically:
- The relationships between patient numbers and embryo numbers need to be clearly defined
- The flow of cases through different stages needs to be accurately represented
- Numbers should match those presented in the main text and tables
- Editorial Correction
Line 557 "Table 77. mothers" should be "77 mothers" for clarity and correct formatting.
These revisions will enhance the manuscript's accuracy, consistency, and overall clarity. Please ensure all numerical data are consistent across the text, tables, and figures.
Please address these points in your revision to improve the scientific rigor and readability of the manuscript.
